# Predicting Compressive Strength and Hydration Products of Calcium Aluminate Cement Using Data-Driven Approach

**DOI:** 10.3390/ma16020654

**Published:** 2023-01-09

**Authors:** Sai Akshay Ponduru, Taihao Han, Jie Huang, Aditya Kumar

**Affiliations:** 1Department of Materials Science and Engineering, Missouri University of Science and Technology, Rolla, MO 65409, USA; 2Department of Electrical and Computer Engineering, Missouri University of Science and Technology, Rolla, MO 65409, USA

**Keywords:** calcium aluminate cement, XGBoost model, analytical model, compressive strength, phase assemblage

## Abstract

Calcium aluminate cement (CAC) has been explored as a sustainable alternative to Portland cement, the most widely used type of cement. However, the hydration reaction and mechanical properties of CAC can be influenced by various factors such as water content, Li_2_CO_3_ content, and age. Due to the complex interactions between the precursors in CAC, traditional analytical models have struggled to predict CAC binders’ compressive strength and porosity accurately. To overcome this limitation, this study utilizes machine learning (ML) to predict the properties of CAC. The study begins by using thermodynamic simulations to determine the phase assemblages of CAC at different ages. The XGBoost model is then used to predict the compressive strength, porosity, and hydration products of CAC based on the mixture design and age. The XGBoost model is also used to evaluate the influence of input parameters on the compressive strength and porosity of CAC. Based on the results of this analysis, a closed-form analytical model is developed to predict the compressive strength and porosity of CAC accurately. Overall, the study demonstrates that ML can be effectively used to predict the properties of CAC binders, providing a valuable tool for researchers and practitioners in the field of cement science.

## 1. Introduction

Concrete is the most widely produced-and-used material globally. While the incessant development of global infrastructure (e.g., rapidly growing metropolises and mega-cities) ensures that demand for concrete is ever-increasing, the production of Portland cement (PC) presents considerable energy consumption (≈11 EJ/year [1]) and environmental impact- (≈9% of global CO2 emission is attributed to the production of PC [2,3,4]) related challenges. Calcium aluminate cement (CAC) has been explored as a sustainable alternative to PC [5,6,7]. The main chemical phases of CACs are calcium aluminate (CA)* and mayenite (C_12_A_7_), whereas gehlenite (C_2_AS) and calcium di-aluminate (CA_2_) are the minor phases [8,9], where C = CaO, A = Al_2_O_3_, H = H_2_O and S = SiO_2_. During the production of PC, the calcination of limestone is a significant contributor to CO_2_ emissions, accounting for approximately 60% of the total CO_2_ emissions [10,11]. The lime content of PC typically ranges from 50 to 60%_mass_, whereas the lime content of CAC is usually between 20 to 30%_mass_ [8,12]. As a result, the manufacturing of CAC emits approximately half the amount of CO_2_ emissions compared to the manufacturing of PC. To be specific, the production of 1 g of CAC releases approximately 0.29 g of CO_2_, which is approximately 47% less than the CO_2_ emissions associated with the production of PC [6,10]. In addition to environmental benefits, CAC becomes even more appealing when we consider its rapid strength achievement. This is on account of the rapid hydration of CAC, because of which 1-day and 7-day strength of the CAC binder are equivalent to 7- and 28-day strengths of their PC counterparts [13,14,15].

During the hydration process of CAC, various intermediate, metastable phases are formed. These phases are called CAH_10_, C_2_AH_8_, and C_4_AH_x_, where “x” can be 11, 13, or 19, depending on the relative humidity [16,17]. These phases later transform into C_3_AH_6_ (hydrogarnet) and AH_3_ (gibbsite) [8]. The formation of these metastable phases are temperature dependent. At low temperatures (below 15 ⁰C), only CAH_10_ is formed [8]. As the temperature increases, C_2_AH_8_ starts to appear in the CAC binder [18]. At around 40 °C, C_2_AH_8_ is the main hydration product formed, along with alumina gel [19]. When the temperature reaches 60 °C, C_3_AH_6_ and AH_3_ are formed without forming any metastable phases [8,18,19]. If the CAC contains silica, C_2_ASH_8_ (straetlingite) may also be formed [9]. Straetlingite is a strength-providing phase that can improve the compressive strength of the CAC. The phase conversion process has a significant effect on the compressive strength of CAC. As the low-density phases (i.e., C_2_AH_8_, CAH_10_, and C_4_AH_x_) transform into the high-density phase (i.e., C_3_AH_6_), the porosity of the cement increases and its compressive strength decreases [14,20].

The conversion of metastable hydrates into stable ones is influenced not only by temperature but also by other factors, such as the water-to-cement ratio and the presence of admixtures. The water-to-cement ratio plays a significant role in this process. At high water-to-cement ratios, the conversion of CAH_10_ into C_3_AH_6_ is complete, but excess water is left over from the reaction, leading to an increase in porosity and a decrease in strength [12,21,22]. On the other hand, at low water-to-cement ratios, there is insufficient water available for CAH_10_ to fully react and convert into C_3_AH_6_, resulting in a more significant amount of CAH_10_, leading to a decrease in porosity and an increase in mechanical strength [8,23]. Several studies [8,24,25,26] have found that inorganic salts can significantly impact the hydration reaction of CAC. It is generally agreed upon that lithium salts accelerate the hydration of alumina-based cementitious materials, with Li_2_CO_3_ being the most common and effective accelerator [26,27]. The addition of Li_2_CO_3_ reduces the time-to-set of CAC (i.e., the time until initial hydrates form) and increases strength development at early ages [25]. However, the use of inorganic salts also has a negative effect, as they decrease the setting times of high alumina-containing CACs and hinder the development of mechanical properties at later ages [25,28]. According to Luong et al. [29], accelerating admixtures only affect the hydration kinetics of CAC initially, prolonging the precipitation of CAH_10_ and C_2_AH_8_. However, once the hydration products start to form, accelerating admixtures do not affect the rate of CAC hydration [30].

Based on the above discussion, it is clear that the mixture design and processing parameters have a significant impact on the mechanical properties and hydration products of CAC. To better understand the influence of these parameters on cementitious materials, previous studies [31,32,33,34,35,36,37,38] have developed several analytical models to predict the compressive strength. These models help researchers to quantitatively understand the effects of different mixture designs and processing parameters on the performance of CAC. The selective equations for analytical models are listed in Table 1. Where: $f_{c}^{\prime }$ is compressive strength; *V_i_* is volume of the component *i*; *w*, *c*, and *a* represents water, cement, and air content, respectively; α is degree of hydration; t is cement curing age; τ is reference curing age; and *A*, *B*, and *n* are coefficients.

The first-generation analytical models to predict the compressive strength of cementitious material were proposed by Feret et al. [37] and Abrams et al. [38], which reinforced the effect of the water-to-cement ratio on the consequent compressive strength. However, these two models did not account for other vital factors (i.e., aggregate content, admixture, curing condition, and concrete age) that are widely known to affect mechanical property development. In 1946, Powers et al. [31] amended the relationship presented by Feret et al. [37] by introducing the degree of hydration and gel-to-space ratio terms. Karni et al. [32] refined the gel-to-space ratio parameter by relating it to the total reacted paste volume. In 2000, Tango et al. [33] simplified the Powers’ equation and initially applied time (i.e., concrete age) to the equation. Around the same time, Popovics et al. [34,39] incorporated air and C_3_S content into Abrams et al.’s model [38]. More recently, multi-factor models—as developed by AL-Shukaili et al. [35] and Gavela et al. [36]—were established to account for the complex cement mixture designs (i.e., water-to-cement ratio; percentage of steel fiber; cement content; aggregate content; and aspect ratio). In particular, Gavela et al. [36] developed a sigmoidal function model to predict the compressive strength of concrete via similar mixture design and processing methods in relation to the water-to-cement ratio and curing time.

However, those models cannot accurately predict the compressive strength of CAC due to several knowledge gaps that exist in the aforesaid analytical models. One of the main reasons is that most of these models were initially developed for PC, which has different strength-providing phases and hydration mechanisms compared to CAC. Next, coefficients in analytical models are not generic, which means that they need to be calibrated each time they are used to predict the compressive strength of a new CAC. Additionally, due to the incomplete understanding of CAC, some theories and parameters cannot be included in analytical models, resulting in lower prediction accuracy. Therefore, an advanced model is required to produce the compressive strength of CAC in a high-fidelity manner.

Machine learning (ML) techniques, a data-driven approach, are a promising tool to achieve reliable predictions of the compressive strength of CAC. ML is an emerged approach used by many studies related to cementitious materials. Previous studies have employed ML models to predict the compressive strength of cementitious materials. Artificial neural networks (ANN) have been used to predict the compressive strength of self–compacting concrete containing bottom ash and self-compacting concrete after exposure to high temperatures [40,41]. Dantas et al. and Duan et al. [42,43] have applied the support vector machine model to predict the compressive strength of concrete made from recycled aggregate. Particle swarm optimization adaptive network-based fuzzy interference and Genetic algorithm adaptive network fuzzy interference models have been developed to predict the compressive strength of alkali-activated concrete made from steel slag [44]. The gray model, a combination of ML and theoretical models, is developed to accurately predict the shear capacity of reinforced concrete [45]. Mangalathu et al. [46] have explored the ability of ML models in predicting the failure mode of reinforced concrete shear walls. Our previous studies [47,48,49,50] have also shown that ML is powerful enough to predict and optimize the mechanical properties of PC and alkali-activated cement. Although ML demonstrates outstanding performance in predicting the properties of various cementitious materials, it has not been utilized to predict the compressive strength of CAC.

Overall, the objective of this study is to provide an accessible method to predict the properties of CAC binders. First, thermodynamic simulations are utilized to obtain phase assemblages at given ages. The XGBoost model is then employed to learn correlations between properties (i.e., compressive strength, porosity, and phase assemblage) and mixture designs, and subsequently produce reliable predictions of those properties. A closed-form analytical model is then developed to predict the compressive strength and porosity based on the variable importance evaluated by the XGBoost model. The analytical model can help end-users who cannot use ML to predict the compressive strength and porosity of CAC binders before starting cumbersome experiments. To the authors’ best knowledge, this is the first study to develop ML and analytical models to predict time-dependent compressive strength and phase assemblages of CAC in relation to mixture designs.

## 2. Modeling Methods

### 2.1. Thermodynamic Model

Several studies [17,51,52] have shown that thermodynamic modeling, coupled with accurate and complete thermodynamic databases, can accurately simulate phase assemblages of hydrates and anhydrates based on the chemical composition of cement. The following paragraphs describe the parameters for Gibbs Free Energy Minimization Software (GEMS) [53,54] simulation and the details of the GEMS simulation itself. The accuracy and completeness of the physicochemical properties of precursors and products used in thermodynamic modeling are critical to the quality of the results. These properties can typically be found in the literature and thermodynamic databases. In this study, the thermodynamic data for aqueous species and solids are taken from the PSI-GEMS thermodynamic database, while the solubility products for cement minerals are taken from the Cemdata 14.01 database [55,56]. An extended Debye-Huckel equation [57] is used to calculate the activity coefficients of the aqueous species.

The performed GEMS simulations utilized the initial mixture design parameters of a given binder (i.e., as inputs at 25 °C. The GEMS calculated the volume of all reactants and hydration products with respect to the degree of hydration. Figure 1 is an example of phase assemblage obtained from the thermodynamic simulation for a CAC binder with a 0.3 water-to-cement ratio. Based on this figure, the hydration process stops at 42% because of a lack of sufficient water to fully hydrate the CAC. This phase assemblage can help researchers to discover the hidden correlations between hydration products and properties at different hydration levels. For example, when 10% CAC reacts with water, it forms 28.571%_vol_ unreacted CAC, 6.1784%_vol_ C_4_AH_19_, 0.7046%_vol_ straetlingite, 1.2367%_vol_ gibbsite, 0.0418%_vol_ hematite, and 0.2745%_vol_ magnetite. The compressive strength would be low because the volume of strength providing phases (i.e., C_4_AH_19_ and straetlingite) are low. The phases shown in the figure are in agreement with findings in prior studies. Lothenbach et al. [16] and Barnes et al. [18] have discovered that C_4_AH_19_, straetlingite, and gibbsite are the main hydration products at 25 °C.

### 2.2. XGboost Model

The XGBoost model [58] is an advanced *classification and regression tree* (CART) model. It uses the gradient boosting technique [59] to combine a set of weak base learners into strong learners through additive functions. The XGboost model is designed to be parallelizable, computationally efficient, and to prevent overfitting. Similar to the conventional CART model, the XGBoost also grows through binary split in a hierarchical fashion. However, the gradient boosting technique allows XGBoost to utilize the objective function (Equation (1)), a combination of cost function and regularization (Ω) at each node, and to train new trees with residual errors (Equation (2)) from previous trees. Therefore, the final output compensates for errors produced by each weak learner.
(1)$$Obj=\underset{i}{\sum }L\left({\hat{y}}_{i},y_{i}\right)+\underset{k}{\sum }Ω\left(f_{k}\right)\;\mathrm{where}\;Ω\left(f\right)=\gamma T+\frac{1}{2}\lambda \omega^{2}$$
(2)$${\hat{y}}_{i}^{\left(t\right)}=\overset{t}{\underset{k=1}{\sum }}f_{k}\left(x_{i}\right)={\hat{y}}_{i}^{\left(t-1\right)}+f_{t}\left(x_{i}\right)$$

Here, *L* is the loss function that measures differences between prediction ${\hat{y}}_{i}$ and target $y_{i}$; ${\hat{y}}_{i}^{\left(t\right)}$ is the prediction at t-th iteration; Ω*(f_k_)* penalizes the complexity of tree *f_k_*; T is the number of leaves of tree *f_k_*; ω is the leaf weights; γ is the complexity of each leaf, and λ is the vector of scores on leaves. Second-order Taylor expansion is then applied to optimize the objective function in the general setting [60], as shown in Equation (3).
(3)$$Obj^{\left(t\right)}=\overset{n}{\underset{i=1}{\sum }}[g_{i}f_{t}\left(x_{i}\right)+\frac{1}{2}h_{i}f_{t}^{2}\left(x_{i}\right)]+Ω\left(f_{t}\right)\;=\sum_{j=1}^{T}[(\sum_{i\in I_{j}}g_{i})\omega_{j}+\frac{1}{2}\left(\sum_{i\in I_{j}}h_{i}+\lambda )\omega_{j}^{2}\right]+\gamma T$$

Here, $g_{i}$ and $h_{i}$ are the first and second, respectively, a derivate of the loss function (L); *I_j_* represents all instances at leaf node *j*. The objection functions before and after the split (Equations (4) and (5)) are compared to determine the effectiveness of a certain split. *I* is the instance set before the split. *I_L_* and *I_R_* are instance sets of left and right nodes after the split. The comparison can be applied to every possible split. If the decision tree’s performance improves after the split, this modification will be accepted; otherwise, the split will be terminated.
(4)$$Obj_{leaf}=-\frac{1}{2}\frac{{\left(\sum_{i\in I_{j}}g_{i}\right)}^{2}}{\sum_{i\in I_{j}}h_{i}+\lambda }+\gamma$$
(5)$$Obj_{split}=-\frac{1}{2}\left(\frac{{\left(\sum_{i\in I_{L}}g_{i}\right)}^{2}}{\sum_{i\in I_{L}}h_{i}+\lambda }+\frac{{\left(\sum_{i\in I_{R}}g_{i}\right)}^{2}}{\sum_{i\in I_{R}}h_{i}+\lambda }\right)+2\gamma$$

Due to the abovementioned architectures, the XGBoost model has several unique features. First, the objective function effectively eliminates overfitting, and thus the XGBoost model converges at a global minimum quickly after a few iterations and maintains the level constantly [61]. Moreover, the objective function used in XGBoost automatically penalizes individual trees, which allows each tree to have a different number of leaves and increases the diversity between trees. This helps prevent overfitting and improves the overall performance of the model. Another unique feature of XGBoost is the use of *shrinkage*, which reduces the influence of individual trees and nodes on future trees. This ensures that the model is able to develop rational input-output correlations and improve its accuracy. Furthermore, XGBoost includes a randomization parameter known as subsampling [58], which decorrelates individual trees. This helps to prevent overfitting and improves the model’s ability to generalize to new data. Overall, the XGBoost model is easy to implement and only requires manual adjustment of a few hyperparameters, such as the *shrinkage* and the *number of iterations*. This makes it a popular choice among data scientists and machine learning practitioners. In this study, the optimal *shrinkage* and the *number of iterations* were 0.2 and 300.

## 3. Database

In this study, the experimental data porosity and compressive strength of CAC were obtained from Matusinovic et al.’s study [62]. The CAC was supplied by Istra Cement International, Pula, Croatia, a part of the Heidelberger Zement Group. The CAC had 40.2%_mass_ CaO, 39.0%_mass_ Al_2_O_3_, 11.7%_mass_ Fe_2_O_3_, 4.3%_mass_ FeO, and 1.9%_mass_ SiO_2_. The principal mineral phase was monocalcium aluminate (CA), with C_12_A_7_, C_6_AF_2_, and C_2_S as minor phases. The Li_2_CO_3_ used was a commercial Analar grade reagent [62]. In mixture designs of CAC binders, water-to-cement ratios were 0.2, 0.25, and 0.3; the Li_2_CO_3_ contents were 0, 0.001, 0.003, 0.005, 0.007, and 0.01%_mass_. The ages of CAC binders were 1, 2, 3, 4, 5, 6, 7, 8, 9, 24, 72, and 168 h. The compressive strength measurement was conducted based on ASTM C349 [63] and ASTM C109 [64]. The compressive strength for each binder was calculated as the average of measurements of triplicate specimens. All experiments were conducted consistently under the abovementioned experimental conditions. In Matusinovic et al.’s study [62], total water (TW) and bound water (BW) in each binder were measured. This information was used to calculate porosity in CAC binders. The TW and BW were calculated by the mass difference between the crushed sample and the sample after removing all water or free water through ignition. The quantity of TW and BW was expressed per 100 g of ignited material. The total porosity (%) is defined as the fraction of the cement paste volume filled with free water, as shown in Equation (6) [24,62,65]. Here, ρ_H2O_ is the density of water (g/cm^3^); ρ_S_ is average density of acetone–dried CAC paste (g/cm^3^; including the hydrates and the fraction of non-reacted cement). After determining the porosity of the CAC binder, the degree of hydration of CAC was estimated. To obtain accurate phase assemblages, it is necessary to carefully specify the degree of hydration of CAC in thermodynamic simulations. The GEMS simulations are performed in a range of degrees of hydration of CAC. Afterwards, the porosity in the phase assemblage is compared to the porosity calculated using Equation (6). If the two porosities match, a straight line is drawn on the phase assemblage figure, which indicates the degree of hydration of the CAC and the volume of the anhydrates and hydrates.
(6)$$P=\frac{\left(\mathrm{TW}-\mathrm{BW}\right)∕\rho_{H_{2}O}}{\frac{100+\mathrm{BW}}{\rho_{s}}+\frac{\mathrm{TW}-\mathrm{BW}}{\rho_{H_{2}O}}}$$

In this study, compressive strength, porosity, and phase assemblage of CAC binders in relation to mixture design and age are consolidated into a single database (shown in Appendix A), which consists of 171 unique data-records. All data-records from Matusinovic et al. [62] are included in the database. This is because the database is adopted from the literature, and there is no possible mechanism to distinguish data-records that were measured accurately from ones that are erroneous. Therefore, for all data-records, we simply assume that all data-records were obtained from proper measurements and were reported accurately, and, therefore, there is no need to “sanitize” the database. The training dataset comprises 75% random-selected data-records from the parent database, and the remaining data-records are utilized as a testing dataset. Through the training dataset, the ML model discovers underlying correlations between mixture design and property. The testing dataset is used to validate the performance of the ML model. Overall, the parent database comprised of three input variables and six outputs. The inputs included mixture designs of the CACs: water-to-cement ratio (unitless); Li_2_CO_3_ content (%_mass_); and cement age (hour). The outputs are compressive strength (MPa), porosity (%_Vol_), C_4_AH_19_ content (%_Vol_), Straetlingite content (%_Vol_), Gibbsite content (%_Vol_), and Solid content (%_Vol_). Other hydrate phases (i.e., Hematite and Magnetite) were not investigated because they were minor phases and provided little-to-no strength. The curing condition was not considered as an input parameter because all the experiments were conducted under identical curing conditions. The statistical parameters pertaining to the database are itemized in Table 2. Four statistical parameters—Pearson correlation coefficient (*R*); coefficient of determination (*R^2^*); root mean squared error (*RMSE*); and mean absolute error (*MAE*)—are used to evaluate the prediction performance of the XGBoost model.

## 4. Results and Discussion

### 4.1. Machine Learning Prediction

This study presents the prediction results of the compressive strength and hydration products of CAC using an XGBoost model. The hyperparameters of the XGBoost model were optimized using a 10-fold cross-validation [66] and grid-search method [67,68], ensuring a robust correlation between the input and output data, accounting for any outliers in the database and eliminating bias and variance influences.

Figure 2 demonstrates the predictions of the compressive strength and porosity against measured/estimated values. Figure 3 exhibits predictions of the volume fraction of C_4_AH_19_, straetlingite, gibbsite, and solid content against phase assemblages estimated by thermodynamic simulations. The results of both the training and testing datasets are shown in the figures. Four statistical parameters pertaining to the prediction results on the testing dataset are shown in Table 3. In Figure 2 and Figure 3, the XGBoost model produces predictions in a high-fidelity manner, where most data-records in both testing and training datasets are located between 10% error lines. In Table 3, the *R* and *MAE* of compressive strength are 0.94 and 5.58 MPa. The marginal error from predictions is in a reasonable range, where the standard deviation of compressive strength measurement ≈ 5 MPa [69]. The *R^2^* values for the predictions of five phase assemblages are larger than 0.90, which implies that the XGBoost model can produce reliable predictions of phase assemblages of CAC. It is not a surprise that the XGBoost model yields reliable predictions for CAC because several studies [61,70,71] have already demonstrated that the XGBoost model produces excellent predictions of various properties of cementitious materials. Such reliable performance contributes to XGBoost’s advanced structures. When each tree grows, the split at each node must be evaluated by the objective function, which allows the model to remove redundant leaves and perform the optimal split. In addition, the cost function helps with the elimination of overfitting and underfitting. In the model, shrinkage, a vital parameter, ensures that the model does not converge at local minimums. Furthermore, the randomized subsampling characteristic guarantees that the structure of each tree is independent from one another. Lastly, only two hyperparameters, *shrinkage rate* and *number of iterations*, are required to be adjusted manually. However, the adjustment may be sluggish and compromise accuracy [66]. To avoid such problems, the grid-search method and the 10-fold CV method, are implemented to optimize the two hyperparameters.

Overall, Figure 2 and Figure 3 demonstrate that the XGBoost model can predict the mechanical properties and phase assemblages of CAC binders as a function of mixture design and cement age. Though outside of the scope for this study, to build upon the results presented in this section, the next step would be to employ an optimization approach to formulate mixture designs while satisfying user-imposed thermodynamic and mechanical criteria, even without a comprehensive understanding of the underlying nonlinear relationships.

Next, the XGBoost model is utilized to evaluate the influence of each input variable on compressive strength and porosity. Figure 4 ranks the influence (importance) of input variables in descending order according to their abilities to change the compressive strength and porosity. This rank can be used as a guideline to develop analytical models by assigning more weight to influential parameters and removing insignificant parameters.

As shown in Figure 4, as expected, cement age is the most influential factor for compressive strength and porosity. This is because as time passes, CAC reacts with water to form C_4_AH_19_, gibbsite, and straetlingite, which monotonically reduces the porosity and increases the compressive strength. The water-to-cement ratio demonstrates a more substantial influence than Li_2_CO_3_ content. This is because the CAC cannot form hydration products that provide strength without sufficient water; nevertheless, excessive water reduces the connectivity between solids and increases porosity, which, in turn, significantly reduces the compressive strength [8,17]. Li_2_CO_3_ is evaluated to be a substantially less critical parameter compared with the other two parameters. Although Li_2_CO_3_ can accelerate the degree of hydration of CAC, its effect on mature properties—compressive strength and porosity—at late ages can be neglected [29].

### 4.2. Analytical Model Development

This section presents an analytical model development based on outcomes from the XGBoost. Based on Figure 4, the relative importance of cement age is determined to be more significant than the water-to-cement ratio and Li_2_CO_3_ content for this dataset. Thus, during the development of the analytical model, cement age is given more weight than the other inputs—which are assigned less, but equal weights due to their relative lack of influence on compressive strength and porosity. Owing to only three input parameters (water-to-cement ratio, Li_2_CO_3_ content, and age), Gavela’s model [36] is selected as a baseline model to develop the simple, closed-form analytical model for CAC. In e Gavela’s model, water-to-cement ratio and age are major input variables. AL-Shukaili’s model [35] is then utilized to elucidate Li_2_CO_3_ content, as a first-order input, in the analytical model. The analytical model that predicts the compressive strength and porosity of CAC is shown in Equation (7). The porosity of cement is directly related to compressive strength; therefore, the same equation is utilized to predict porosity. Here, *C_i_* is the constant coefficient (unitless); r is the water-to-cement ratio (unitless); Li is the Li_2_CO_3_ content (%_mass_), and t is the cement age (hour). By using a nonlinear, gradient-descent scheme [49,72] and Nelder-Mead multidimensional simplex algorithm [73,74], the constant and coefficients are optimized, as shown in Table 4.
(7)$$ompressive\;strength/Porosity=\left|\left(C_{1}+C_{2}*r\right)*e^{\frac{-c_{3}}{t}}+C_{4}*Li+C_{5}t^{C_{6}}+C_{7}\right|$$

Figure 5 shows the predicted compressive strength and porosity produced by the analytical model against measured values. The statistical parameters pertaining to prediction accuracy are shown in Table 5. In general, the analytical model produces predictions of the compressive strength and porosity of CAC with reasonable accuracy. The *R^2^* and *MAE* of compressive strength predictions are 0.87 and 7.67 MPa, respectively, and the *R^2^* and *MAE* of porosity predictions are 0.82 and 4.24%_vol_, respectively. In Figure 5, it is worth pointing out that the predictions of compressive strength under 20 MPa exhibit higher deviation between predicted and measured values than any other predictions owing to variations in Li_2_CO_3_ content. Within the dataset of interest, the compressive strength under 20 MPa were measured from the CAC binder at early ages. The Li_2_CO_3_ content nonlinearly affects the hydration reaction at early ages, substantially increasing the variation of compressive strength and decreasing prediction performance. A potential solution that improves the prediction accuracy is to utilize a more extensive database or extend the current one further. Inclusion of more data-records into the database will enhance its volume and diversity, which, in turn, can be utilized to optimize the analytical model further and enhance prediction performance. To the authors’ best knowledge, this is the first study to develop a closed-form analytical model to predict the compressive strength and porosity of CAC.

## 5. Conclusions

CAC, which has traditionally been used in refractory applications, has gained popularity as a CO_2_-efficient alternative to PC. However, the unpredictable nature of CAC has limited its widespread adoption. Hence, there is a need to understand the composition-property relationships in CAC. Recently, machine learning (ML) has been used to uncover non-linear correlations between composition and properties in composite materials. By using mixture design attributes such as cement age, water-to-cement ratio, and Li_2_CO_3_ content as inputs, ML can predict not only the porosity and compressive strength but also the phase assemblages of hydrated CAC. However, ML techniques may not be universally accessible. As an alternative, a novel analytical model has been developed to predict the compressive strength and porosity of CAC.

In this study, the compressive strength and porosity of CAC was obtained from previous studies. Based on the mixture design, thermodynamic simulations were used to determine the phase assemblages of CAC at different degrees of hydration. The XGBoost model was used to predict the compressive strength, porosity, and phase assemblages of CAC in relation to the mixture design and cement age. The results showed that the XGBoost model can produce reliable predictions of the properties of CAC. Additionally, the model was able to evaluate the impact of different input parameters on the compressive strength and porosity of CAC. This information was used to guide the development of a closed-form analytical model that can predict the compressive strength and porosity of CAC. Analytical approaches can be more desirable because they do not require any programming background to perform predictions. The optimized analytical model produced predictions of the compressive strength and porosity of CAC with good accuracy. In conclusion, the prediction accuracy of both the XGBoost and analytical models could be improved by using a more extensive and a more diverse dataset. This study marks an important step towards developing machine learning models to predict the properties of CAC. In the future, a larger and more diverse database of CAC may be applied to the XGBoost model. By learning the input-output correlations from this new database, the XGBoost model will be able to easily predict the properties of CAC with different mixture designs and processing parameters. Additionally, the XGBoost model has the potential to optimize the mixture design of CAC to achieve specific target properties.

## Figures and Tables

**Figure 1 materials-16-00654-f001:**
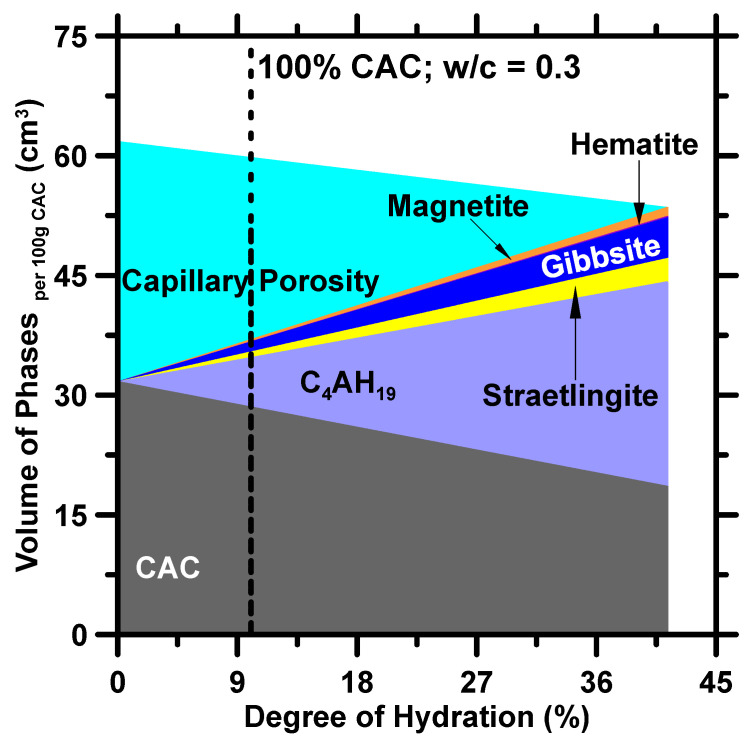
Phase assemblage of CAC estimated through GEMS at various degrees of reaction. The dashed line indicates the phase assemblage at 10% reacted CAC based on the degree of hydration estimated from compressive strength. After 41.3% of CAC reacted, the hydration reaction was terminated due to insufficient water.

**Figure 2 materials-16-00654-f002:**
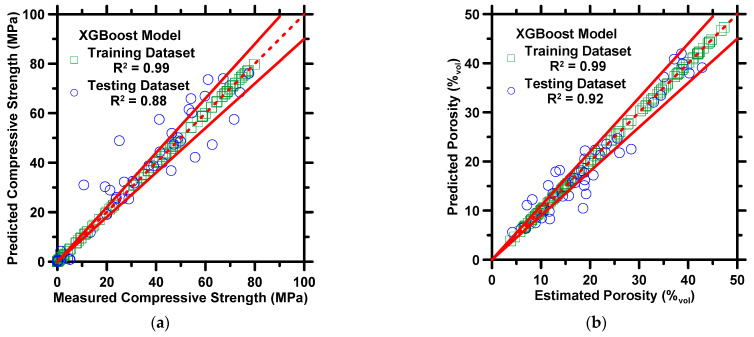
The XGBoost model’s predictions of: (**a**) compressive strength, and (**b**) porosity against experimental measurements from Matusinovic et al. [62]. The coefficient of determination (*R^2^*) is shown in the legend. The dashed line represents the line of ideality, and the solid lines represent a ±10% error bound.

**Figure 3 materials-16-00654-f003:**
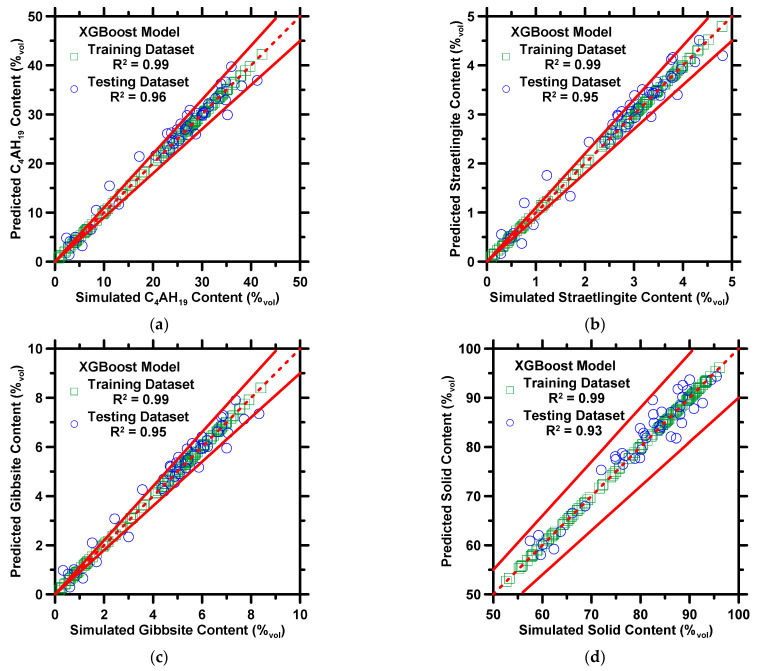
The XGBoost model’s predictions of: (**a**) C_4_AH_19_ content, (**b**) straetlingite content, (**c**) gibbsite content, and (**d**) solid content against phase assemblages derived from thermodynamic simulations. The coefficient of determination (*R^2^*) is shown in the legend. The dashed line represents the line of ideality, and the solid lines represent a ±10% error bound.

**Figure 4 materials-16-00654-f004:**
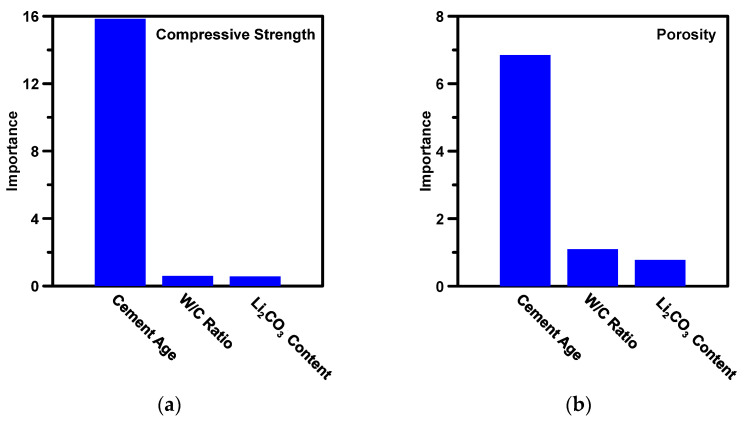
The importance of input parameters contributing to (**a**) compressive strength; and (**b**) porosity of CAC. Parameters are listed as a descent trend in relation to their decreasing influence on the property.

**Figure 5 materials-16-00654-f005:**
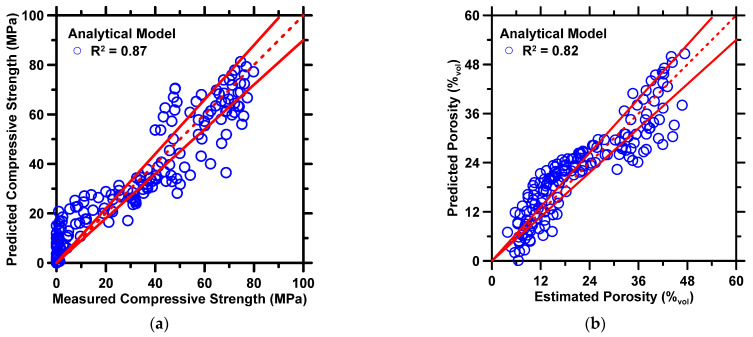
The analytical model’s predictions of: (**a**) compressive strength, and (**b**) porosity against experimental measurement from Matusinovic et al. [62]. The coefficient of determination (*R^2^*) of each prediction is shown in the legend. The dashed line represents the line of ideality, and the solid lines represent a ±10% error bound.

**Table 1 materials-16-00654-t001:** Analytical models used to predict the compressive strength of cementitious materials.

Analytical Model	Reference
$f_{c}^{\prime }=A{\left(\frac{V_{c}}{V_{c}+V_{w}+V_{a}}\right)}^{B}$	Feret et al. [37]
$f_{c}^{\prime }=\frac{A}{B^{w/c}}$	Abrams et al. [38]
$f_{c}^{\prime }=A{\left(X\right)}^{B}=A{\left(\frac{0.66\alpha }{\frac{w+V_{a}}{c}+\alpha }\right)}^{B}$	Powers et al. [31]
$f_{c}^{\prime }=\left(A+B\frac{w}{c}\right)e^{\left[-\tau /t^{n}\right]}$	Gavela et al. [36]

**Table 2 materials-16-00654-t002:** Summary of minimum, maximum, mean, and standard deviation of the CAC database population related to three inputs and six outputs (bold). The database consists of 171 unique data records.

Attribute	Unit	Min.	Max.	Mean	Std.Dev.
Water-to-Cement Ratio	Unitless	0.20	0.30	0.25	0.04
Li_2_CO_3_ Content	%_mass_	2.55	10.91	3.89	1.17
Cement Age	Hour	1	168	30.42	51.62
**Compressive Strength**	MPa	0	79.80	33.08	26.65
**Porosity**	%_Vol_	3.84	47.34	21.35	11.96
**C_4_AH_19_ Content**	%_Vol_	0.74	42.23	21.78	11.44
**Straetlingite Content**	%_Vol_	0.09	4.80	2.48	1.30
**Gibbsite Content**	%_Vol_	0.16	8.40	4.34	2.27
**Solid Content**	%_Vol_	52.64	96.13	78.65	11.96

**Table 3 materials-16-00654-t003:** Four statistical parameters (i.e., *R*, *R^2^*, *MAE*, and *RMSE*) evaluating the performance of XGBoost on predictions of compressive strength, porosity, C_4_AH_19_ content, straetlingite content, gibbsite content, and solid content.

	*R*	*R^2^*	*MAE*	*RMSE*
**Compressive Strength**	**Unitless**	**Unitless**	**MPa**	**MPa**
	0.9386	0.8809	5.577	8.088
**Porosity**	**Unitless**	**Unitless**	**%_Vol_**	**%_Vol_**
	0.9578	0.9173	2.353	2.967
**C_4_AH_19_ Content**	**Unitless**	**Unitless**	**%_Vol_**	**%_Vol_**
	0.9789	0.9582	1.693	2.155
**Straetlingite Content**	**Unitless**	**Unitless**	**%_Vol_**	**%_Vol_**
	0.9757	0.9520	0.2060	0.2598
**Gibbsite Content**	**Unitless**	**Unitless**	**%_Vol_**	**%_Vol_**
	0.9757	0.9520	0.3839	0.4553
**Solid Content**	**Unitless**	**Unitless**	**%_Vol_**	**%_Vol_**
	0.9655	0.9322	2.205	2.716

**Table 4 materials-16-00654-t004:** List of coefficients of the analytical model for the compressive strength and porosity of CAC.

Compressive Strength	C1	82.159	C2	−41.801	C3	3.632
	C4	1788	C5	1.631	C6	0.274
	C7	−15.422				
Porosity	C1	2.610	C2	55.324	C3	−0.488
	C4	−710	C5	35.542	C6	−0.258
	C7	−16.192				

**Table 5 materials-16-00654-t005:** Four statistical parameters (i.e., *R*, *R^2^*, *MAE*, and *RMSE*) evaluated the prediction performance of the analytical model on compressive strength and porosity.

	*R*	*R^2^*	*MAE*	*RMSE*
**Compressive Strength**	**Unitless**	**Unitless**	**MPa**	**MPa**
	0.9351	0.8745	7.666	9.673
**Porosity**	**Unitless**	**Unitless**	**%_Vol_**	**%_Vol_**
	0.9075	0.8236	4.244	5.065

## Data Availability

The database used in this study is provided in the Appendix A.

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
