# Peer review of "Predicting Compressive Strength and Hydration Products of Calcium Aluminate Cement Using Data-Driven Approach"

_materials, 2023, doi:10.3390/ma16020654_

Round 1

Reviewer 1 Report

The topics of the article are important, the issue of predicting the strength of CAC composites is still problematic. The authors' approach can be considered interesting and in line with current trends. However, I have a number of doubts. First of all, the authors used one rather broad database, but only one and therefore limited (1 type of cement, narrow w/c variation). They did not validate the model using other databases or their own research. This is a considerable flaw in the article, which at the very least needs to be addressed in the discussion. Secondly, I am not so optimistic about the model's agreement with the research results - the differences at the level of individual determinations are large and, for practical reasons, difficult to accept. Thirdly, the presented model requires the determination of a number of coefficients that depend, among other things, on the composition of the CAC cement. Thus, the model is not general, but specific to the cement used. Fourth, the mode does not take into account curing conditions. 

Without denying the possibility of publication of the article, the above comments and concerns of the authors should be taken into account in the discussion in the article.

Reviewer 2 Report

The manuscript presents a numerical study for predicting the compressive strength of Calcium aluminate cement (CAC) through machine learning. The influence of input parameters on the compressive strength and porosity of CAC is studied through XGBoost model and a closed-form analytical model is derived based on the results of this study.

The paper needs a revision before reconsideration for publication in the journal Materials.
The following comments should be considered by the authors when revising the paper:
1)    Some sentences need revision, in terms of grammar structure or mistakes such as in the introduction “Resulting in approximately half the amount of CO2 emission when compared to the manufacturing of PC” should be linked with the previous sentence (maybe the full stop should be replaced with comma). Another example is in Eq. 7, in which “Porsity” should be corrected with “Porosity”. Other typos are noted in Figure 5 (b) (both in the “Analytcial Model” and in the x-axis title “Estimiated Porosity”). Please double check again the entire manuscript for typos.
2)    It is advised to move some text from the introduction to specific subsections in the following parts of the manuscript, instead of reporting all these details in the introduction. As an example, the detailed part about the different hydration process of CAC compared to Portland cement (PC) could be discussed in a dedicated section, whereas in the introduction only the main focus of this paper and the novelty in comparison to the state of the art should be reported. This would improve the readability of the paper.
3)    To widen the generality of the discussion, the bibliography (and the introductory comments) should be integrated with contributions in the field of machine learning that employ mechanical concepts in the area of concrete structures in general, not only compressive strength determination. Among these contributions, particularly relevant are those to predict the shear strength capacity of reinforced members (including RC columns and beams) and those to predict he seismic behavior of reinforced concrete shear walls:
Quaranta, G., De Domenico, D., & Monti, G. (2022). Machine-learning-aided improvement of mechanics-based code-conforming shear capacity equation for RC elements with stirrups. Engineering Structures, 267, 114665.
Mangalathu, S., Jang, H., Hwang, S. H., & Jeon, J. S. (2020). Data-driven machine-learning-based seismic failure mode identification of reinforced concrete shear walls. Engineering Structures, 208, 110331.
4)    The reliability of the study results for other researchers and for experimental data other than those selected in this study strongly relies on the homogeneity and consistency among the selected specimens. Although reference is made to [60] for the database selection, some clarifications could be reported to justify the homogeneity of the selected experimental results taken from the literature. The accuracy of the prediction models is significantly influenced by the completeness of the data, quantity, quality, and distribution of the input parameters. These aspects should be critically discussed by the authors.
5)    How do the authors think that the results of this study can be useful for other researchers and, above all, to practitioners involved in design tasks?
6)    Which is the reliability of Eq. 7? Did the authors quantitatively evaluate it? This is very important for design purposes.
7)    Do the authors intend to share their developed codes? Is there a public repository?
8)    How can the analysis presented here (in terms of methods and results) be applicable to other mix compositions? Additionally, more future research should be included in conclusion part.

Round 2

Reviewer 1 Report

Article can be accepted/

Reviewer 2 Report

The authors have satisfactorily addressed the previous Reviewer’s concerns by modifying and integrating the manuscript. The revised version of the manuscript is recommended for publication in Materials.